

# Increased precipitation enhances soil respiration in a semi-arid grassland on the Loess Plateau, China

Yutao Wang[1], Yingzhong Xie[1], Gillian Rapson[2], Hongbin Ma[1], Le Jing[1], Yi Zhang[1], Juan Zhang[1] and Jianping Li[1]

[1] School of Agriculture, Ningxia University, Yinchuan, Ningxia, China
[2] School of Agriculture and Environment, Massey University, Palmerston North, New Zealand

## ABSTRACT

**Background**. Precipitation influences the vulnerability of grassland ecosystems, especially upland grasslands, and soil respiration is critical for carbon cycling in arid grassland ecosystems which typically experience more droughty conditions.

**Methods**. We used three precipitation treatments to understand the effect of precipitation on soil respiration of a typical arid steppe in the Loess Plateau in north-western China. Precipitation was captured and relocated to simulate precipitation rates of 50%, 100%, and 150% of ambient precipitation.

**Results and Discussion**. Soil moisture was influenced by all precipitation treatments. Shoot biomass was greater, though non-significantly, as precipitation increased. However, both increase and decrease of precipitation significantly reduced root biomass. There was a positive linear relationship between soil moisture and soil respiration in the study area during the summer (July and August), when most precipitation fell. Soil moisture, soil root biomass, pH, and fungal diversity were predictors of soil respiration based on partial least squares regression, and soil moisture was the best of these.

**Conclusion**. Our study highlights the importance of increased precipitation on soil respiration in drylands. Precipitation changes can cause significant alterations in soil properties, microbial fungi, and root biomass, and any surplus or transpired moisture is fed back into the climate, thereby affecting the rate of soil respiration in the future.

## INTRODUCTION

Recent changes in global temperatures and precipitation patterns have occurred due to the increase in greenhouse gases (*Gao et al., 2016*). Temperatures are expected to gradually increase in most parts of the world and extremes are anticipated to become more frequent (*Garrett et al., 2006*). Global warming is expected to cause atmospheric water vapor to increase significantly and affect the hydrological cycle (*O'Gorman & Schneider, 2009*), impacting global precipitation patterns and causing regional precipitation changes (*Pall, Allen & Stone, 2006*). The intensity of precipitation events is expected to increase, and extreme precipitation events will occur more frequently according to the forecast of the Intergovernmental Panel on Climate Change (*IPCC, 2001*).

Corresponding author
Jianping Li, lijianpingsas@163.com

Previous studies have indicated that changes in precipitation affect the dynamics of the terrestrial carbon cycle and terrestrial carbon pools (*Ahlström et al., 2015*; *Felton, Knapp & Smith, 2019*; *Frank et al., 2015*; *Wu et al., 2011*). Water is a driving factor for chemical and biological processes in ecosystems, including plant survival, photosynthesis and respiration, heterotrophic respiration (*Gerten et al., 2008*; *Zhang et al., 2019a*), soil nutrient dynamics (*Yuan et al., 2017*), terrestrial ecosystem functioning (*Wu et al., 2009*), and microbial activity, diversity, and respiration (*Classen et al., 2015*; *Felton, Knapp & Smith, 2019*; *Frank et al., 2015*). Moreover, changes in global precipitation patterns exert profound effects on the nature of vegetation (*Gao et al., 2016*), especially in arid and semiarid regions where water is the main limiting factor for plant growth (*Jing et al., 2010*; *Knapp et al., 2002*).

Global emissions of $CO_2$ from soil are considered to be one of the largest causes of flux in the global carbon cycle and small changes in soil respiration may have large impacts on atmospheric $CO_2$ concentrations (*Schlesinger & Andrew, 2000*). Soil $CO_2$ flux is sensitive to higher temperatures, since root exudates, root mycorrhizae, plant detritus, and other part of the plant have different temperature sensitivities to fluctuations in soil $CO_2$ levels (*Boone et al., 1998*), which themselves strongly impact the terrestrial carbon cycle (*Fischlin et al., 2007*; *Frank et al., 2015*). However, we still have a poor understanding of the response of soil respiration as related to climate change (*Fernandez et al., 2006*).

Carbon fluxes have recently been studied in alpine meadows and show that only underground biomass and soil moisture have a direct effect on soil respiration (*Geng et al., 2012*). Studies on tropical forests have shown that higher elevation decreases in root and litter samples resulted in increases in soil respiration, with microbial respiration more closely related to soil moisture levels (*Zimmermann et al., 2010*). In arid temperate grasslands, soil respiration has a positive response to extreme precipitation events, affecting the ecosystem's carbon cycle (*Thomey et al., 2011*). However, there are few studies on semi-arid ecosystems which account for about 15% of the terrestrial area of the globe (*Huang et al., 2015*). They are especially sensitive to precipitation changes (*Niu et al., 2019*). Often occurring in semi-arid areas, grasslands play a key role in the carbon cycle (*Li et al., 2017*; *Poulter et al., 2014*; *Zhang et al., 2019b*), while being very sensitive to large-scale climate change (*Feng, An & Wang, 2006*).

The upland Loess Plateau in northwestern China is a critical transition zone for semi-arid ecosystems in China (*Zhao, Chen & Ma, 2014*). It is predicted that temperature and precipitation will increase significantly in this area (*Zhao, Chen & Ma, 2014*), where we conducted field work to simulate increased and decreased precipitation exploring the factors affecting soil $CO_2$ flux. Our study has great scientific and practical implications for the effects of precipitation on soil respiration and plant productivity and their roles in regional and global terrestrial carbon cycles. We sought to evaluate the differences in soil properties, plant biomass, and microbial diversity at different soil depths using different precipitation treatments, to determine the main factors affecting soil respiration in the study area.

 

## MATERIALS & METHODS

### Study sites

The study was conducted at the Agriculture Experimental Station of Ningxia University , Yinchuan Province of China, in Yunwu Mountain of Guyuan, Ningxia (106°21′E−106°27′E, 36°10′N−36°17′N). The climate of the area is temperate, continental monsoon. The annual average precipitation is about 439 mm and varied from 282 mm in 1982 to 706 mm in 2013. More than 50% of the annual precipitation occurs in the summer months (June to August). The average annual temperature is 7.2 °C, and varied from 5.3 °C in 1984 to 8.7 °C in 2013. The average monthly minimum temperature for the coldest month (January) was −7.2 °C, and the average monthly maximum temperature for the warmest month (July) was 19.6 °C. The annual evaporation is 1,300–1,640 mm, and the annual duration for sunshine can reach 2500 h, with a frost-free period of 112–140 days. The annual potential evapotranspiration is 1,625 mm (meteorological data from 1981 to 2017 is from the National Meteorological Administration of China). The soils are grey-cinnamon and dark loessial, as classified by the Chinese soil classification system (*National Soil Survey Office, 1993*). The vegetation is typical steppe and the main plant species are *Stipa bungeana*, *Artemisia gmelinii*, *Stipa grandis*, *Artemisia frigida*, *Potentilla acaulis* and *Agropyron michnoi* (*Wang et al., 2020*).

### Experimental design

Our research site was located in a semi-arid natural grassland that was left ungrazed for 19 years. The study area was at 2,077 m, with a 7−10° slope, and a south-facing, sunny aspect. Annual precipitation in 2019 was 592 mm, which was 20% higher than average. We set three blocks with three 6 × 6 m plots in each block, to make a total of 9 plots. Each trio of plots down the gradient was considered to be a block (Fig. 1). According to local multi-year meteorological data from the study area, the mean maximum and minimum precipitation levels were about 50% and 150% of the average annual precipitation. A Rain shelter was set up on each block. Each rain-shelter was fixed to the ground by steel pillars, and transparent polyethylene plates were fixed in "V" shapes to intercept precipitation and channel it off the plot using the natural slope of the mountain, while forming a stable and well-ventilated structure (*Afreen & Singh, 2019*). Rain shelters intercepted half of the natural precipitation to form a reduced precipitation treatment (R50). The intercepted water was piped to an adjacent plot to form an increased precipitation area (R150). The remaining three plots were the controls (R100). Snow was collected from the rain shelters after each snowfall (R50) and was sprinkled evenly into the R150 plots. Plastic barriers were used to prevent surface runoff or leakage of soil moisture between plots. Each barrier was buried at a depth of 1.1 m and projected 10 cm above ground. Our study ran from May 2017 to May 2019.

### Environmental factors

We collected field data in July 2019 which corresponded to the annual period of peak biomass. Three soil sample replicates were collected at depths of 0–9.9 cm, 10–19.9 cm, and 20–30 cm in each plot, after the litter was discarded. The soil samples were separated

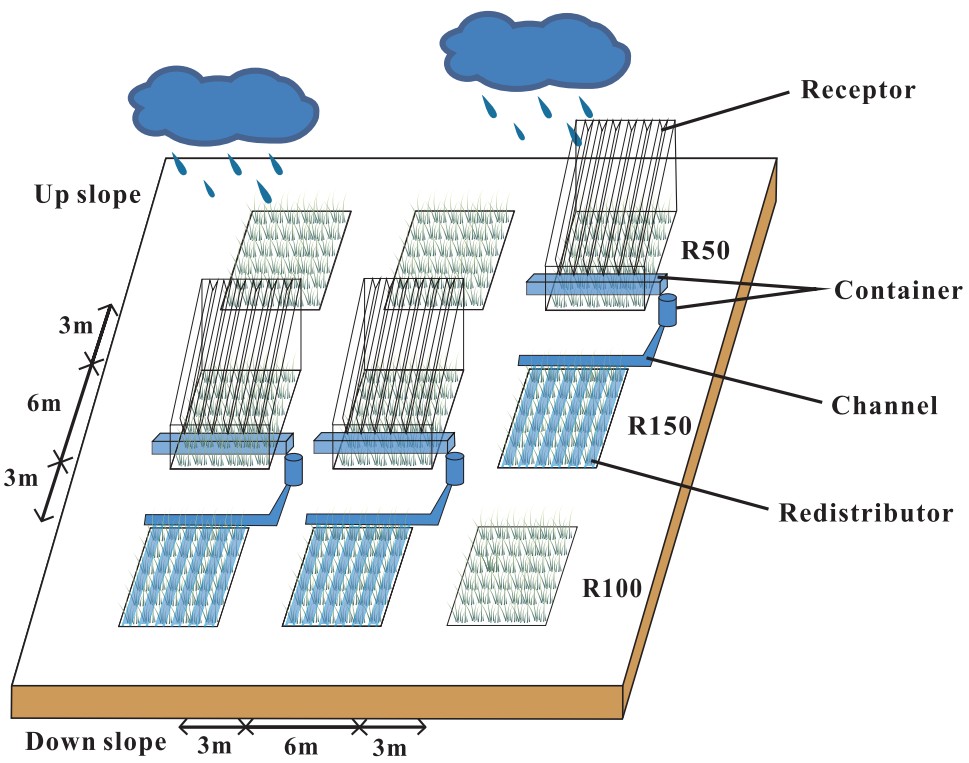

**Figure 1** **Rain-shelter construction and layout of the subplots at the study area.** Three precipitation treatments were applied: R50 (= 50% of ambient precipitation), R100 (ambient) and R150 (= 150% of ambient precipitation).

into two parts: one part was kept moist to determine the microbial diversity of the soil, and the other was air-dried for measurement of soil properties.

Soil organic carbon (SOC) was measured by potassium dichromate-sulfuric acid digestion, with ammonium ferrous sulfate titration. Total nitrogen (TN) was determined using an Elementar analyzer (Elementar, Vario EL III, Germany). Total phosphorus (TP) was measured using Olsen's method (*Bremner & Mulvaney, 1982*). Soil pH was measured using a PHS-3C pH Meter in a 1:5 ratio of fresh soil: water slurry (Huakeyi, Beijing, China).

A 1 m² quadrat was randomly selected in each subplot to determine the plant biomass. The litter was raked and bagged and the shoots of the plants were cut at ground level. Root biomass (RB) was sampled to 30 cm in three intervals of equal depth and the soil carefully brushed off the roots. All plant samples were dried at 65 °C in an oven for 75 h and then weighed.

Soil microbial diversity was determined based on the Illumina HiSeq sequencing platform of the Majorbio Cloud Platform (http://www.majorbio.com). The bacterial primer was 338F_806R and the fungal primer was ITS1F_ITS2R. Sobs' and Shannon's indices were used to indicate the alpha diversity of bacteria and fungi. Coverage index, as defined by Good (*Good, 1953*) indicates the percentage of operational taxonomic units

(OTUs) sampled in a microbial community (i.e., recovered per sample) as a percent of all OTUs found on the site (*Chao, 1984*; *Lemos et al., 2011*).

## Measurements of soil respiration

Soil respiration (release of $CO_2$) was measured every 14–16 days by the LI-8100A portable gas exchange system from April 2019 to October 2019 (LiCor, Lincoln NE, USA; chamber 8100-103, diameter of 20 cm). Polyvinyl chloride (PVC) collars were set permanently in place in the soil one week before the first measurement to minimize soil disturbance. The height of each collar was 12 cm and the above-ground height was 3 cm, with the soil surface area and volume within the collar being 317.8 $cm^2$ and 953.4 $cm^3$, respectively. Five collars were placed randomly in each plot, giving a total of 45 collars. The above-ground parts of the plants inside the collar were removed before taking each flux reading and the roots were left in place (*Afreen & Singh, 2019*). Soil fluxes were measured about every 16 days, between 9 am and 1 pm, based on weather conditions. The flux from each collar was measured for 100 s. Soil moisture was measured at a depth of 5 cm with a GS-1 Licor sensor, and the temperature was measured concurrently at a depth of 10 cm using the Licor sensor 6000-09 TC. To avoid pseudo-replication, the five values per plot were averaged for each variable to get a single datum for each measurement timepoint.

## Statistical analysis

Statistical analysis was conducted using IBM SPSS (IBM, Chicago, USA). One-way ANOVA was used to process the aboveground and underground biomass of plants under different precipitation treatments. A two-way ANOVA was used to process root biomass, soil nutrient content, soil pH, and the microbial diversity index under different precipitation treatments and different soil depths. Microbial diversity was calculated by Mothur (Version v.1.30). Origin (Origin Lab 2017; Microcal, MA, USA) was used for figures. The different dates formed temporal pseudoreplication, so the significance was explored using nlme (https://svn.r-project.org/R-packages/trunk/nlme) in R (*R Core Team, 2013*) for soil temperature, soil moisture, and soil $CO_2$ flux (using the format lme: flux ∼Precipitations * Date, random = ∼1|plot, weights = varIdent (form = ∼1|Date)). The varIdent function in package Predictmeans v1.0.2 (https://www.rdocumentation.org/packages/predictmeans) was used to allow each timepoint to have a different variance. Package ggplot2 (https://www.rdocumentation.org/packages/ggplot2) was used for correlations of soil $CO_2$ flux and all other factors to filter out some variables. For example, if the correlation between Soil SOC and Soil TN was high, we only selected one of these variables, and then performed sPLS analysis (Partial Least Squares regression) (http://mixomics.org/methods/spls/). By contrast, if the correlation between the two was poor, we performed sPLS on both variables. Finally, we considered including soil moisture, RB 0-19.9, soil pH, soil fungi in a sPLS analysis to select variables influencing the predictive model of soil $CO_2$ flux. Stepwise regression was used to model of the main factors affecting soil respiration.

## RESULTS

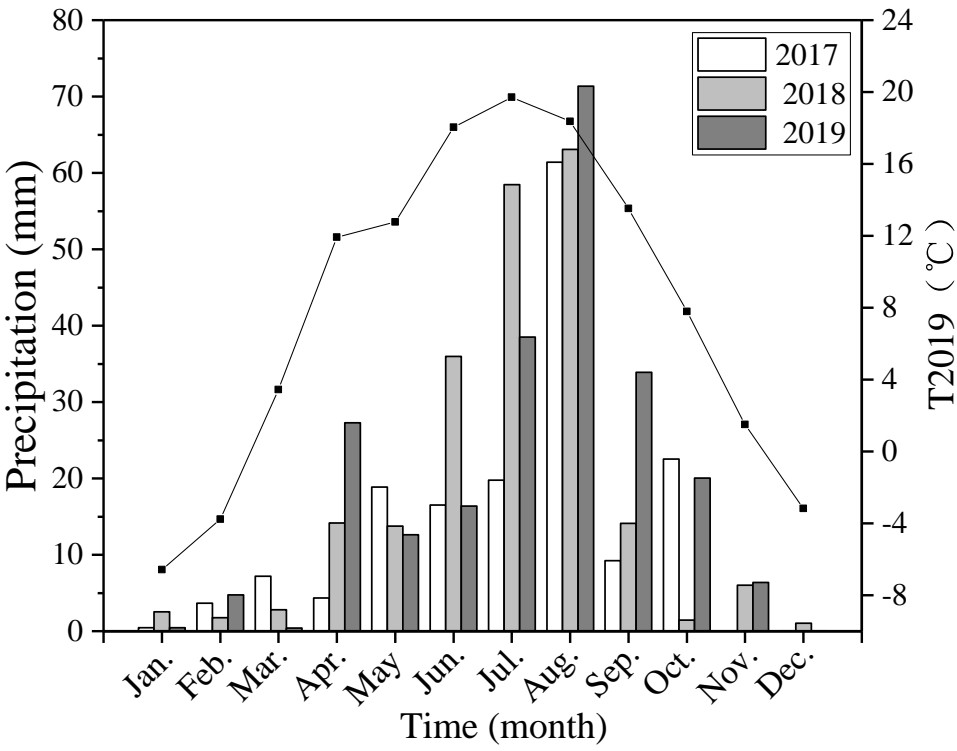

**Figure 2** **Monthly precipitation at the study site from 2017 to 2019 (bars) and average monthly temperature (T2019) in 2019 (black line and points).** Meteorological data were obtained from the National Meteorological Administration of China.

## Precipitation and temperature during the trial

The annual precipitation was 420 mm, 550 mm, and 592 mm in 2017–2019. In 2018 and 2019 precipitation was 30% higher than the average level of precipitation for the last 40 years. Precipitation in 2019 from April to October (the growing season) was 562 mm, which was about 95% of the average annual precipitation (Fig. 2). The average annual air temperature was 7.9 °C in 2019, which was 9% higher than the average temperatures from 1980 to 2019. The highest temperature was 19.7 °C in July, and the lowest was −6.6 °C in January in 2019 (Fig. 2).

## Soil properties

Two-way ANOVA results showed that, except for TP, soil properties had significant differences under precipitation treatments and at different soil depths ($P < 0.05$). Precipitation and soil depth gave the only significant interaction in SOC. The rest of the soil properties had their highest value under R100 in soil 0–30 cm deep (Table 1), with the exception of the soil TP. SOC was 23% greater under R50 as the depth of the soil layer was greater, followed by R100 (18%) and R150 (14%). SOC at R50 increased just 1% in the topsoil (0–9.9 cm) and decreased about 6% in soil depths of 20–30 cm, compared with R100, while R150 had same value with R100 in 20–30 cm soil depths but decreased about 3% in the 0–9.9 cm soil band. As the depth of the soil layer increased, soil pH gradually

Wang et al. (2021), *PeerJ*, DOI 10.7717/peerj.10729

**Table 1  Soil nutrient content and soil pH in different precipitation treatments and soil depths.**

| Variables | R50 | | | R100 | | | R150 | | | P-Value | | |
|---|---|---|---|---|---|---|---|---|---|---|---|---|
| | SD1 | SD2 | SD3 | SD1 | SD2 | SD3 | SD1 | SD2 | SD3 | Pre | SD | Pre*SD |
| TN (g kg-1) | 2.3 ± 0.1 a | 2.1 ± 0.1 b | 1.9 ± 0.1 b | 2.4 ± 0.0 a | 2.1 ± 0.0 b | 2.1 ± 0.1 b | 2.3 ± 0.1 a | 2.0 ± 0.1 b | 2.0 ± 0.0 b | **0.041** | **0.000** | 0.253 |
| SOC (g kg-1) | 8.1 ± 0.1 a | 6.1 ± 0.3 bc | 6.2 ± 0.2 c | 8.0 ± 0.0 a | 7.0 ± 0.1 b | 6.6 ± 0.3 bc | 7.7 ± 0.3 a | 6.8 ± 0.2 bc | 6.6 ± 0.3 bc | **0.045** | **0.000** | **0.049** |
| TP (g kg-1) | 0.7 ± 0.0 | 0.7 ± 0.0 | 0.7 ± 0.0 | 0.7 ± 0.0 | 0.8 ± 0.0 | 0.7 ± 0.0 | 0.7 ± 0.0 | 0.7 ± 0.0 | 0.7 ± 0.0 | 0.231 | 0.687 | 0.213 |
| pH | 7.8 ± 0.1 ab | 7.8 ± 0.1 ab | 7.9 ± 0.1 ab | 7.8 ± 0.1 ab | 8.0 ± 0.2 a | 8.1 ± 0.1 a | 7.7 ± 0.1 b | 7.9 ± 0.1 ab | 8.0 ± 0.2 ab | **0.022** | **0.003** | 0.626 |

**Notes.**

Pre, Precipitation; SD, soil depth; SD1, soil depth 0–9.9 cm; SD2, soil depth 10–19.9 cm; SD3, soil depth 20–30 cm; Pre*SD, Precipitation*soil depth.

Values are mean ± standard error ($n = 3$).

Means in a row without a common superscript letter differ ($P < 0.05$) as analyzed by two-way ANOVA.

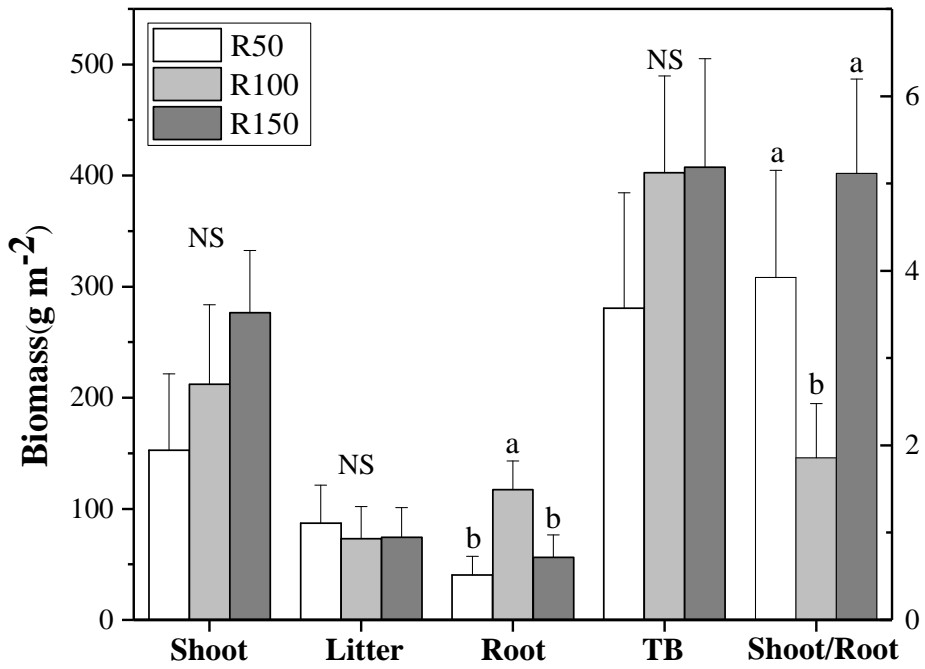

**Figure 3** **Biomass of shoot, litter, roots, total biomass (TB) and shoot/root ratio for each precipitation.** Different letters within each tissue type show significant differences ($P < 0.05$) between precipitation treatments according to one-way ANOVA (mean ± standard error, $n = 3$). Shoot, litter, roots and TB use the left Y axis, and shoot/root the right axis.

increased. Soil TN and SOC were highest in the topsoil (0–9.9 cm), and decreased with soil depth.

## Biomass of shoot, litter and root

There were no significant differences in shoot, litter, and total biomass (TB= sum of shoots, litter, and RB 0-30) under different precipitation treatments according to one-way ANOVA's results. Nevertheless, shoot biomass was greatest in R150, and was lower at lower precipitation. Litter biomass was greatest in R50. By contrast, root biomass (RB) 0–30 cm showed significant differences between precipitation treatments according to one-way ANOVA , with R100 having the most root biomass, and R150 and R50 being significantly lower by 52% and 65% when compared with R100, respectively (Fig. 3A). The total biomass under R50 was lower than for treatments R100 and R150. The shoot/root ratio (Aboveground biomass/RB), was largest in R150, and was significantly lower by 64% in R100 ($P < 0.05$) (Fig. 3B).

Different precipitation treatments and soil depths caused significant differences in root biomasses according to two-way ANOVA. All values were lower at greater soil depths (Fig. 4). The highest value of root biomass appeared at 0–9.9 cm in R100; values in R50 and R150 at 0–9.9 cm were significantly lower by 68% and 57%, respectively. Both increased precipitation and reduced precipitation significantly reduced the root biomass in the topsoil (0–9.9 cm) (Fig. 4).
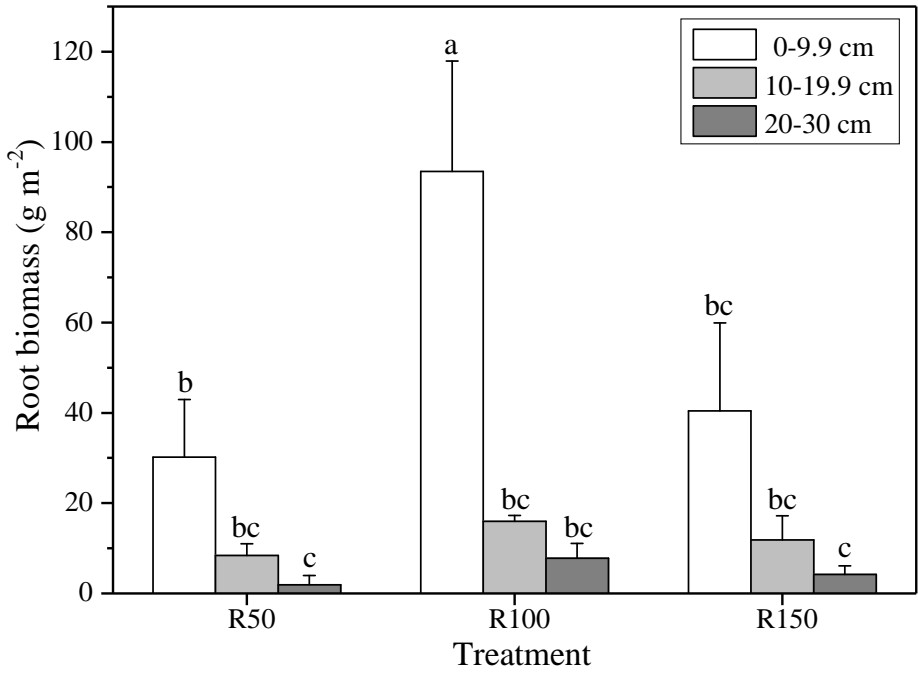

**Figure 4 Root biomass at different soil depths in three precipitation treatments.** Different letters show significant differences ($P < 0.05$) between precipitation treatments and soil depths according to two-way ANOVA (mean ± standard error, $n = 3$).

## Microbial richness and diversity

More than 95% of operational taxonomic units (OTUs) found in the microbial community at the study site were present in each soil sample examined. There was no significant difference in soil microbial richness via Sobs' index and diversity via Shannon's index under different precipitation treatments and for the precipitation*soil depth interaction. However, there was a significant difference in soil microbial richness and diversity with soil depth, and the value was greatest at 0–9.9 cm (Table 2). The Sobs' index and Shannon's index of bacteria and fungi gradually decreased at greater soil depth.

## Soil moisture, soil temperature and soil CO$_2$ flux

Soil moisture, soil temperature and soil CO$_2$ flux showed significant differences between precipitation treatments over the entire experimental period (April–October) (Table 3). Measurements over the growing season also differed significantly for all variates and were significantly affected by precipitation levels ($P < 0.05$) (Table 3).

Soil moisture and CO$_2$ flux were greater at higher levels of precipitation while soil temperature was lower (Table 4). The response of soil moisture to the decreased precipitation treatment ($-37.2\%$ of R100) was greater than to increased precipitation (3.9% of R100). Mean soil CO$_2$ flux under R100 was higher by 38.9% than that for R50, while for R150 the increase relative to R100 was only 8.3% (Table 4).

All variables (soil moisture, soil temperature and soil CO$_2$ flux) were strongly influenced by seasonality (Table 3). Soil moisture was highest in April and May, then dropped to rise

**Table 2  Alpha diversity of bacteria and fungi under different precipitation treatments and soil depths.**

| Variables | R50 | | | R100 | | | R150 | | | P-Value | |
|---|---|---|---|---|---|---|---|---|---|---|---|
| | SD1 | SD2 | SD3 | SD1 | SD2 | SD3 | SD1 | SD2 | SD3 | Pre | SD |
| B_Sobs | 2529 a | 2183 b | 2037 bc | 2615 a | 2157 b | 2024 bc | 2530 a | 2222 b | 1880 c | 0.46 | **0.00** |
| B_Shannon | 6.7 ab | 6.3 d | 6.2 d | 6.7 a | 6.4 cd | 6.2 d | 6.6 abc | 6.4 bcd | 6.2 d | 0.44 | **0.00** |
| B_Cov(%) | 96.0 | 96.4 | 96.5 | 95.8 | 96.4 | 96.6 | 95.8 | 96.3 | 96.9 | 0.82 | **0.01** |
| F_Sobs | 725 ab | 528 abcd | 491 bcd | 755 a | 458 cd | 414 cd | 632 abc | 490 bcd | 352 d | 0.11 | **0.00** |
| F_Shannon | 4.3 ab | 3.4 ab | 3.3 ab | 4.5 ab | 4.0 ab | 3.2 b | 4.7 a | 3.8 ab | 3.7 ab | 0.33 | **0.00** |
| F_Cov(%) | 99.8 | 99.8 | 99.9 | 99.8 | 99.9 | 99.9 | 99.9 | 99.9 | 99.9 | **0.02** | **0.00** |

Notes.

B, Bacteria; cov, coverage; F, Fungi.

Community richness (Sobs' index), community diversity (Shannon's index) and community coverage (of OTUs) of bacteria and fungi under different precipitation treatments and soil depths. Different letters show significantly different values between depths within each precipitation treatment.

**Table 3  ANOVAs for soil temperature, soil moisture and soil $CO_2$ flux between precipitation treatments and dates during the whole experimental period.**

| | numDF | denDF | Soil moisture | | Soil temperature | | Soil $CO_2$ flux | |
|---|---|---|---|---|---|---|---|---|
| | | | F-value | P-value | F-value | P-value | F-value | P-value |
| Intercept | 1 | 60 | 822.1 | <0.0001 | 17856 | <0.0001 | 1626.0 | <0.0001 |
| Pre treatments | 2 | 6 | 42.7 | 0.0003 | 5.4 | 0. 046 | 55.9 | 0.0001 |
| Dates | 10 | 60 | 414.5 | <0.0001 | 348.0 | <0.0001 | 156.7 | <0.0001 |
| Pre treatments * Dates | 20 | 60 | 14.8 | <0.0001 | 8.0 | <0.0001 | 9.7 | <0.0001 |

Notes.

num DF,  number of degrees of freedom; den DF,  the number of degrees of freedom associated with the model errors.

**Table 4  Precipitation treatments differences for soil moisture, soil temperature and soil $CO_2$ flux over the whole experimental period.**

| Precipitations | R50 | R100 | R150 |
|---|---|---|---|
| Mean soil moisture (%) | 8.1 ± 2.3 b | 12.9 ± 2.9 ab | 13.4 ± 3.5 a |
| Mean soil temperature (°C) | 19.2 ± 1.3 a | 18.9 ± 1.1 a | 18.7 ± 1.1 a |
| Mean soil $CO_2$ flux ($\mu$mol $CO_2$ m$^{-2}$ s$^{-1}$) | 2.2 ± 0.4 b | 3.6 ± 0.8 ab | 3.9 ± 0.8 a |

Notes.

Values are the mean ± standard error ($n = 3$). There are three precipitation treatments, and each treatment has three replicates. Sampling was conducted 11 times during the growing season. Different letters show significantly different between precipitation treatments ($P < 0.05$).

again in October, giving a "W"-shaped relationship with time. Soil temperatures peaked in summer (June-August). The soil $CO_2$ flux showed an upward trend from April to July, and peaked before decreasing to its lowest levels in October (Table 5).

Soil moisture was typically greatest in the R150 treatment, with the exception of April and September (Fig. 5A). The highest soil temperature values were typically seen in R50 (Table 4), but this situation was reversed in June and July (summer; Fig. 5B), when precipitation was high (Fig. 2). Overall, the soil temperature first rose and then decreased during the growing season.

Soil $CO_2$ flux showed the same trends as soil moisture for most months. The soil $CO_2$ flux was lower at the lower end of the precipitation gradient and was highest at R150 and lowest

**Table 5** Seasonal differences in soil moisture, temperature and $CO_2$ flux across all precipitation treatments.

| Date | Soil moisture (%) | Soil temperature (°C) | Soil CO$_2$ flux ($\mu$mol CO$_2$ m$^{-2}$ s$^{-1}$) |
|---|---|---|---|
| Apr. 23 | 19.7 ± 4.5 | 15.6 ± 1.3 | 3.02 ± 0.8 |
| May 09 | 22.2 ± 4.1 | 10.7 ± 0.9 | 2.58 ± 0.6 |
| May 25 | 2.3 ± 1.4 | 20.8 ± 1.4 | 1.78 ± 0.5 |
| Jun. 09 | 2.8 ± 1.9 | 24.5 ± 2.0 | 2.41 ± 0.6 |
| Jun. 28 | 10.6 ± 3.2 | 20.5 ± 0.9 | 4.33 ± 0.8 |
| Jul. 13 | 11.1 ± 2.9 | 24.7 ± 1.5 | 4.56 ± 0.8 |
| Jul. 27 | 10.2 ± 2.8 | 23.3 ± 1.1 | 4.59 ± 1.1 |
| Aug. 13 | 3.8 ± 1.8 | 22.3 ± 1.2 | 2.69 ± 0.6 |
| Sep. 07 | 9.2 ± 2.7 | 17.4 ± 0.9 | 4.48 ± 0.6 |
| Sep. 21 | 15.5 ± 0.9 | 15.5 ± 0.9 | 3.38 ± 0.7 |
| Oct. 13 | 18.7 ± 5.7 | 13.0 ± 0.9 | 1.59 ± 0.3 |

**Notes.**
Values are the mean ± standard error ($n = 3$).

at R50. However, across the summer months, early May and early July, with enhanced precipitation (R150), the soil $CO_2$ flux was slight lower about 1%–10% , compared to R100 (Fig. 5C). By contrast, precipitation had little effect on soil $CO_2$ flux in the autumnal month of October.

## Modeling soil CO$_2$ flux

Based on the results of the correlation analysis (Fig. S1), sPLS analysis was performed and showed that the factors most closely correlated with soil respiration were soil moisture, root biomass (RB 0-9.9, RB 10-19.9), soil pH, and fungal diversity (Fig. S2). Stepwise regression showed that soil moisture was the only input factor that significantly affected soil $CO_2$ flux ($P < 0.001$, $R = 0.94$, adjusted $R^2 = 0.870$), and all other variables were excluded (Table 6).

## DISCUSSION

Soil organic carbon (SOC), total soil nitrogen (TN) and pH were significantly different in the three precipitation treatments, having their highest values in R100, while both increased and decreased precipitation reduced these values. Nutrients in the soil can accumulate through the degradation of plants, litter, and root secretions (*Qiu et al., 2009*; *Zhang et al., 2016*). Our study showed that different precipitation treatments had no significant impact on biomass of shoots and litter, but there was a significant effect on the root biomass. Increases and decreases in precipitation also significantly reduced root biomass, which was significantly higher in R100, especially in the topsoil (0–9.9 cm). Changes in precipitation may influence a plant's growth as related to the balanced growth hypothesis (*Shipley & Meziane, 2002*). Plants will preferentially distribute any acquired water to the root in semi-arid areas (*Afreen & Singh, 2019*). Therefore, soil properties under different precipitation treatments may be more affected by the root biomass, as root decomposition and roots' secretions will cause differences in SOC, TN and pH. We found

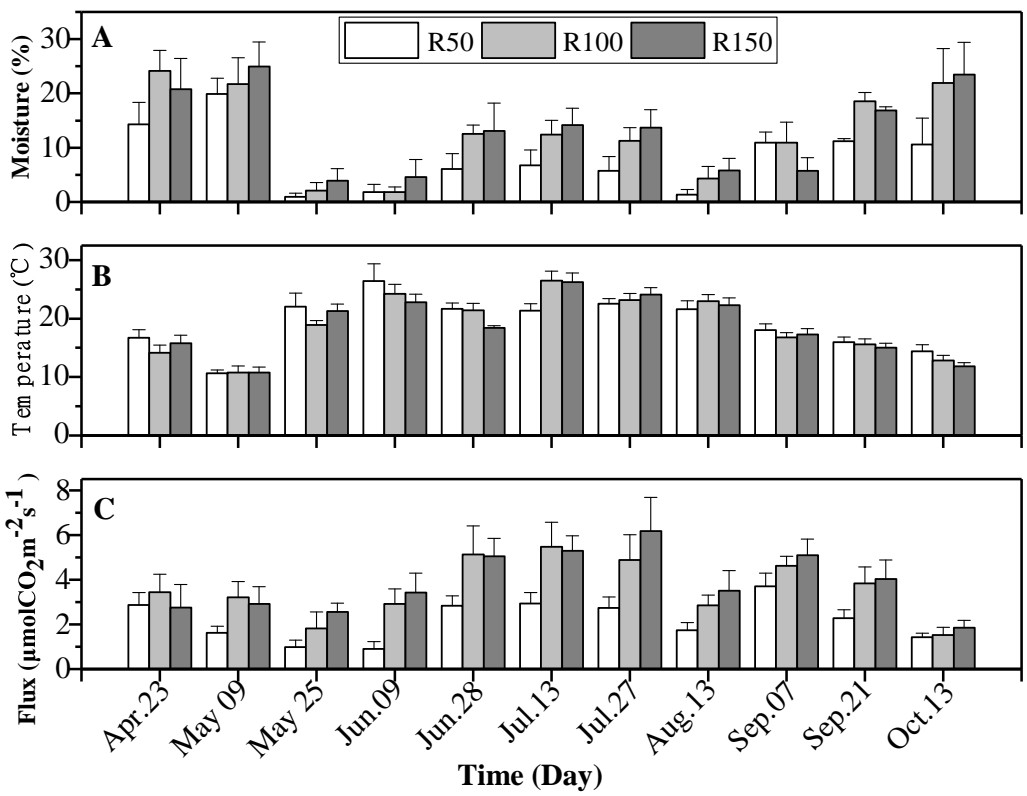

**Figure 5** **Interactions of precipitation treatments and times for soil moisture, soil temperature and soil CO$_2$ flux during the growing season.** Values are the mean $\pm$ standard error ($n = 3$).

**Table 6** **The Predictive model of soil CO$_2$ flux according to Stepwise regression.** It is based on soil moisture and the most informative factors from the July measurements of each of the sets of soil properties, plant factors and soil microbial diversity (Shannon index) in three precipitation treatments.

| Input variable | Excluded variables | R | Adjust R$^2$ | Sig | Durbin-Watson |
|---|---|---|---|---|---|
| Soil moisture | RB 0–9.9 cm, RB 0–9.9 cm, Soil pH, Soil fungal diversity | 0.941 | 0.870 | 0.00 | 2.727 |
| Model | Soil CO$_2$ flux = 0.35 + 0.42 Soil moisture | | | | |

that as precipitation decreased (R50) in the study area, the water requirements for grass growth were no longer met, and root biomass was significantly lower compared with that in R100. As precipitation increased in the summer, the root biomass was higher in R150 than R50 and R100.

The indices for soil microbial diversity and richness were unresponsive to the three precipitation treatments, but were responsive to different soil depths. While previous studies have suggested that water supply is positively correlated with the richness and diversity of the community (*Hawkins et al., 2003*), this rule does not necessarily apply to soil microorganisms (*Bachar et al., 2010*), which may instead be determined by physical isolation of microbial populations (*Treves et al., 2003*). In addition, short-term

precipitation changes of 1 or 2 years have little effect on microbial diversity (*Wang et al., 2020*).

Soil moisture, soil temperature, and soil respiration showed significant seasonal differences under different precipitations throughout the experiment. Soil moisture and soil respiration increased with the precipitation gradient, but soil temperature showed the opposite, downward, trend. Soil temperature may be affected by coverage of surface vegetation (*Kang et al., 2000*), which increases with increasing precipitation in semi-arid environments; the canopy intercepts solar radiation and reduces evaporation, thus lowering soil and air temperatures. Plant biomass is lowest in R50, which had the largest area of bare land and thus probably experienced more evaporation from the bare soil. The higher the soil moisture, the greater the specific heat capacity, so that occurrence of more soil water may lower the soil temperature.

Generally, changes of soil temperature reflected atmospheric temperature in the study area except at very high temperatures, and more precipitation had a weakening effect on the relationship between air and soil temperatures. Temperature regulates soil respiration by changing the rate at which organisms process carbon and nutrients (*Crowther & Bradford, 2013*). Our study showed there was no significant correlation between soil respiration and soil temperature during vigorous plant growth, which may be related to our brief experimental period, only two years. Studies indicate that the temperature sensitivity of soil respiration will decrease under continuous global warming (*Peng et al., 2009*). In addition, an increasing number of studies show that soil respiration responds more strongly to precipitation pulses rather than soil warming pulses in regions limited by water (*Almagro et al., 2009*).

Changes in soil moisture content and precipitation were not closely correlated, but showed high values in spring and autumn and trended in a "W" shape during the growing season. In the spring melting snow replenished the soil moisture, and lower evapotranspiration kept the soil moist. However, high temperature with high evapotranspiration reduced the soil moisture in the study area in summer and limited growth. Precipitation events in water-limited areas usually only affect the topsoil and this water is easily lost by direct evaporation (*Schwinning & Sala, 2004*), resulting in no soil moisture gains. Generally, soil moisture accumulates as air temperature drop and evapotranspiration losses are decreased in autumn (*Felton, Knapp & Smith, 2019*; *Maes & Steppe, 2012*; *Wang et al., 2013*). We found that, under increased precipitation, soil moisture was increased in May and then decreased towards September.

Literature has indicated that other non-negligible factors affecting soil moisture include soil permeability, surface runoff, evaporation, and evapotranspiration (*Wang et al., 2012*). We found that surface run-off could be negated with the use of a plastic barrier with an underground depth of 110 cm and a ground height of 10 cm. A bigger root system can improve soil permeability, aeration, and porosity (*Ozalp, Erdogan Yuksel & Yuksek, 2016*). In our experiment, the greater root biomass caused by increased precipitation may have resulted in improved soil permeability and moisture storage, increasing soil moisture. At the same time, the higher above-ground biomass and canopy coverage under the increased precipitation treatment may have had an inhibiting effect on surface evaporation, which is

intense in arid areas, while the rain-reduction treatment experienced the opposite effect. Soil moisture is one of the main environmental factors affecting soil $CO_2$ flux. Soil moisture affects the physiological performance of microorganisms and nutrient diffusion (*Yuste et al., 2007*), especially in arid areas with limited water conditions (*Emmett et al., 2004*; *Lellei-Kovács et al., 2011*; *Zhang et al., 2010*). A clear threshold effect links soil respiration with soil moisture (*Balogh et al., 2011*), and especially so in arid areas (*Niu et al., 2019*). Our study demonstrated that the total soil $CO_2$ flux in R150 was higher than in R50 throughout the test period, but in April, June, and July when the soil moisture content was higher than in other months, the soil $CO_2$ flux of R50 was almost equal to or even slightly above R150. Spring-time soil moisture replenishment from ice and snow melt caused the soil moisture in R50 to reach the maximum threshold for soil respiration in the study area. Increased precipitation produced little or no effect on soil respiration.

Soil respiration is the comprehensive result of autotrophic respiration mainly based on root activity and heterotrophic respiration related to soil organic matter decomposition (*Wang et al., 2014*). In our experiment, despite other contributory factors including root biomass, pH, and fungal Shannon diversity, multiple regression analysis showed that soil moisture was the best predictor of soil respiration. Stepwise regression showed a significant positive linear correlation between soil moisture and soil respiration throughout the plant growth period, with the strongest correlation at the end of July (peak of summer) ($R^2 = 0.87$). In arid areas, concentrated and heavy precipitation events can greatly stimulate soil respiration (*Liu et al., 2016*), which our results support (*Alwyn et al., 2008*; *Emmett et al., 2004*; *Lellei-Kovács et al., 2011*; *Zhang et al., 2010*). Drought can reduce the diffusion of organic substrates and decrease extracellular enzyme activity, thereby inhibiting root growth and microbial activity and impacting heterotrophic respiration (*Liu et al., 2016*). Our study area was confined to an arid area, which may limit the broader applications of our responses. Future climate change in precipitation may be mostly concentrated in a series of ecosystem responses caused by decreased precipitation and repeated samples should be taken in R50 and at even lower soil moisture levels.

## CONCLUSIONS

In summary, our research showed that both increased and decreased precipitation treatments reduced the root biomass of plants and increased the shoot/root ratio. Precipitation changes caused significant changes in soil moisture, soil temperature, and soil respiration during the whole plant growing season in the study area, but moisture became non-limiting during heavy precipitation episodes at the height of the growing season. Changes in precipitation treatments significantly affected soil nutrients (SOC, TN and pH) across all soil depths, especially in the topsoil. Though significantly affected by soil depth, microbial diversity and richness were not sensitive to precipitation treatments. Overall, in arid grassland ecosystems where water is the limiting factor, the ecological response to changes in soil moisture caused by precipitation changes should be a key research topic to deepen our understanding of soil respiration in the future.

## ACKNOWLEDGEMENTS

We thank all reviewers for their comments on the manuscript. We acknowledge all our team members for their help with field experiments.

### Funding

This study was financially supported by the Research and Development Project of Ningxia (2020BEG03046), Natural Science Foundation of Ningxia (2019AAC03042), and Top Discipline Construction Project of Pratacultural Science (NXYLXK2017A01). The funders had no role in study design, data collection and analysis, decision to publish, or preparation of the manuscript.

### Grant Disclosures

The following grant information was disclosed by the authors:
Research and Development Project of Ningxia: 2020BEG03046.
Natural Science Foundation of Ningxia: 2019AAC03042.
Top Discipline Construction Project of Pratacultural Science: NXYLXK2017A01.

### Competing Interests

The authors declare there are no competing interests.

### Author Contributions

- Yutao Wang performed the experiments, analyzed the data, prepared figures and/or tables, and approved the final draft.
- Yingzhong Xie and Hongbin Ma conceived and designed the experiments, authored or reviewed drafts of the paper, and approved the final draft.
- Gillian Rapson analyzed the data, authored or reviewed drafts of the paper, and approved the final draft.
- Le Jing, Yi Zhang and Juan Zhang performed the experiments, prepared figures and/or tables, and approved the final draft.
- Jianping Li conceived and designed the experiments, performed the experiments, analyzed the data, authored or reviewed drafts of the paper, and approved the final draft.

### Data Availability

The raw measurements are available in the Supplemental Files.

### Supplemental Information

Supplemental information for this article can be found online at http://dx.doi.org/10.7717/peerj.10729#supplemental-information.

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
