# Peer review of "Increased precipitation enhances soil respiration in a semi-arid grassland on the Loess Plateau, China"

_PeerJ, doi:10.7717/peerj.10729_

## Round 0.1 · original submission · Major Revisions

Your manuscript has been reviewed by two experts. Both suggested major revisions and pointed out specific comments for your consideration. I want to add that the English writing should be much improved before resubmission.

Reviewer 1 ·

Basic reporting

This study investigated the impact of precipitation changes on soil respiration in a semi-arid area of the Loess Plateau, China. The authors provided some background information related to this study. But the manuscript was poorly written. There are too many awkward sentences that significantly hinder the readers’ understanding. Some examples are showed below. The authors need to seek help from a native English speaker or a commercial English editing company.

Experimental design

The research is original and fit the aims and scope of the journal. The authors provided some rational in the Introduction. Materials and methods were provided in details in general. But the number of replication (or block) of three was a little bit too low, particularly considering that there were only three precipitation treatments. It was not clear if the transparent polyethylene plates were installed (in reverse V) in the control and R150 plots to avoid the influence of light. For data analysis, the description of correlation analysis (Lines 175-185) was confusing. Ggplot2 seems like a plotting R script. Why was this one used for correlation analysis? I don’t understand why only one variable was selected? I think multiple regression should be used here.

Validity of the findings

The authors provided the results, and results are generally sound. But some conclusion were not fully supported by the results. For example, the authors claim that precipitation influenced aboveground biomass, however, above-ground (shoot and litter) and total biomass were not significantly influenced by precipitation (Fig. 3).

Additional comments

I have a few major concerns with this manuscript. One was the experimental design. The experiment did not use block control, and the plots were not randomly assigned to the plots (Fig. 1). I also feel that the replication number was too small. The second one was data analysis. The authors mentioned ANOVA in the data analysis. But the results were not presented. In the regression analysis, relationship between soil respiration and temperature is usually not linear. Nonlinear regression should be used for soil temperature. The 3rd one was English and presentation. The manuscript needs to be significantly improved. I recommend Reject or Major revision.
Specific comments:
L29: was precipitation change a part of climate change?
L29: have an impact on, or impact (delete on)
L30: In dry ecosystems?
L32: delete (i.e., … grassland)?
L36: across precipitation treatments
L36: soil moisture was influenced by precipitation treatment
L38: there was (Please use past tense to describe current results. In the manuscript, both past tense and present sense are used).
L41-43: replace are with were, is with was
L44-46: run-on sentence. Please revise or separate it into two sentences.
L68-76: The logic flow is not good here. Please move the L72-74 to where soil respiration was introduced above.
L117-119: awkward sentence. Please revise.
L117: No block control was used in this study. The design only considered replication (3 plots for each treatment), but did not apply block control, and did not randomly assigned treatments to plots.
146: GOOD?
L165-166: not clear here.
L169: repeated measure?
L174-180: I’m confused by the data analysis here. It seems that ggplots is R script for plotting. Why used this for correlation (regression) analysis?
L178: why only one variable was selected? I think multiple regression should be applied.
Results: I would suggest to present soil temperature, moisture and soil respiration before Biomass of shoot, litter and root.
Where were results of ANOVA? Treatment effects, depth, date, interactions?
L251: use past tense for the results
L256-257: “So other factors were also analyzed using a linear model”: why?
L259 variables
L259-261: Revise this sentence.
L267-269: Why? So was there a significant treatment effect or not on above-ground biomass?
L280: was
L283: hole? Hold?
L286: The journal of .. has studied?
L333-341: This is background info. Too long here.
L349: what’s Akaike analysis?
L369-371: This is not new.
L372: if not significant, how can the authors say that precipitation can reduce soil nutrient?

Reviewer 2 ·

Basic reporting

The English language should be improved to ensure that an international audience can clearly understand the text.

Experimental design

no comment

Validity of the findings

no comment

Additional comments

Abstract:
The results don't support the conclusion that more precipitation can produce more root biomass. because the highest root biomass was found in R100.

Introduction:
The authors made a review of the recent studies of precipitation on CO2 flux and the need to take a study on grassland in semi-arid areas. however, it failed to descript the knowledge gap and put forward scientific questions or the objectives of this study.
line 97: the font of “—” should be fixed to “-”, and the same comment for line 99, 104, 105, etc.

Material and methods:
line 174: the varIdent function here was used to allow each timepoint rather than precipitation to have a different variance.

Results:
line 196: how did the 23% calculate?
line 198: how does the author define topsoil? 0-9.9 cm or 0-20cm
line 199: what is the meaning of "Soil pH was greater at depth in all three precipitations"
line 204: the text is not consistent with Fig 3A, As showed in the figure there is no significant difference between the three treatments but the authors said "significantly "lower by 45% in R50.
line 208-209:The same comment as line 204
line 211: the Fig.4 showed the Root biomass at different soil depths of three precipitations rather the "volume of roots"
line 215:The same comment as line 204
line 220: be consistent with the Table2, "0-10cm" or "0-9.9cm"
line 256: where is Fig.6?
line 261:no models showed in Table 6 had an AICC<2.

Discussion:
line 269-275: I don't think the authors give a robust explanation of why the R100 had the highest root biomass.
line 280-291: The interpretation may not seem to be robust, As it showed in Table 3 that precipitation showed a significant effect on soil temperature and moisture, however, it is not consistent with the authors’ statement here.
line 290-300: Another reason might that the water has a higher specific heat volume, thus the treatments with more soil water are more likely to have a low soil temperature.
line 328-332: However, more root and above ground biomass may lead to more water was absorbed and evapotranspiration by plants.
line 525: the first letter of the journal name should be capitalized.

Figures and tables:
Fig.1: only the R50 treatments were designed with shelters. how to avoid the shelters' effect on the solar radiation interception, it may differ between R50 and the other two treatments.
Fig.4: Please show the sample size
I don't think it is appropriate to do one-way ANOVA. you have two factors: soil depth and precipitation, so a two-way ANOVA is recommended.
Table 4: Why the sample size was 33?The soil CO2 flux was measured from April 2019 to October 2019 as mentioned in line 151, thus 11 measurements were taken during the experimental period.

---

## Round 0.2 · Minor Revisions

Please make further revisions of your manuscript and pay attention to the comments from reviewer #1. I would be happy to accept you work if the revision is satisfactory.

Reviewer 1 ·

Basic reporting

English and writing have been significantly improved.

Experimental design

Experimental design was adequate.

Validity of the findings

Results are clearly presented mostly.

Additional comments

Comments
The authors made efforts and addressed most of my concerns. There are still a few minor issues with the revision. Please see the specific comments below. I recommend Minor revision.

Specific comments:
(Line numbers in Word file with tracking)
L37: add "of ambient precipitation" after 150%.
L91: delete "the soil CO2 flux"?
L117: We set three blocks with three 6 x 6 m plots in each block, to make ...
L118: change interfere to intercept
L127: after each snowfall
L168: The company of IBM SPSS is IBM, not SPSS now
L173-188: I feel this part is too long. Please shorten this paragraph.
Two references are not found
Fig. 1: The three plots slope can be considered as one block. This study had three blocks (replications).
Table 2. Since only soil depth influenced most variables here, the main effect of soil depth could be presented. There is no need to present the interactive effect of precipitation and soil depth, as the interaction was not significant.
Table 4. Add significant test results to means.

---

## Round 0.3 · Minor Revisions

The science has no major flaws. But the writing and language must be significantly improved to meet the publication standard. Currently, the text has quite a number of errors in grammar, word choice, etc. The authors should seek editing assistance from a fluent speaker or company.

---

## Round 0.4 · accepted · Accept

The revised manuscript is much improved in its language and I suggest acceptance. A minor comment: change the title to "Increased precipitation enhances soil respiration in a semi-arid grassland on the Loess Plateau, China".